# Exercise-Induced Adipose Tissue Thermogenesis and Browning: How to Explain the Conflicting Findings?

**DOI:** 10.3390/ijms232113142

**Published:** 2022-10-28

**Authors:** Yupeng Zhu, Zhengtang Qi, Shuzhe Ding

**Affiliations:** 1The Key Laboratory of Adolescent Health Assessment and Exercise Intervention (Ministry of Education), East China Normal University, Shanghai 200241, China; 2School of Physical Education and Health, East China Normal University, Shanghai 200241, China; 3Sino-French Joint Research Center of Sport Science, East China Normal University, Shanghai 200241, China

**Keywords:** adipose tissue, browning, thermogenesis, exercise, ROS, redox

## Abstract

Brown adipose tissue (BAT) has been widely studied in targeting against metabolic diseases such as obesity, type 2 diabetes and insulin resistance due to its role in nutrient metabolism and energy regulation. Whether exercise promotes adipose tissue thermogenesis and browning remains controversial. The results from human and rodent studies contradict each other. In our opinion, fat thermogenesis or browning promoted by exercise should not be a biomarker of health benefits, but an adaptation under the stress between body temperature regulation and energy supply and expenditure of multiple organs. In this review, we discuss some factors that may contribute to conflicting experimental results, such as different thermoneutral zones, gender, training experience and the heterogeneity of fat depots. In addition, we explain that a redox state in cells potentially causes thermogenesis heterogeneity and different oxidation states of UCP1, which has led to the discrepancies noted in previous studies. We describe a network by which exercise orchestrates the browning and thermogenesis of adipose tissue with total energy expenditure through multiple organs (muscle, brain, liver and adipose tissue) and multiple pathways (nerve, endocrine and metabolic products), providing a possible interpretation for the conflicting findings.

## 1. Introduction

Adipose tissue in the human body is usually divided into white adipose tissue (WAT) and brown adipose tissue (BAT). BAT is rich in mitochondria that contain uncoupling protein 1 (UCP1), which dissipates proton motive force for ATP synthesis to generate heat and regulate body temperature, a process known as non-shivering thermogenesis [1]. BAT was originally thought to be active in newborns or adolescents, sustaining a progressive involution with age. However, functional BAT is still present in adults (such as supraclavicular, neck, vertebral side, mediastinum and kidney) [2]. Another intermediate adipose tissue, brown-like adipose tissue (or beige adipose tissue), can also be generated in adults upon particular stimuli including cold, exercise and β-adrenergic agonists [3,4]. Due to the contribution of BAT to energy expenditure and metabolic regulation, adipose tissue thermogenesis and browning are associated with combating obesity, type 2 diabetes and other metabolic diseases [5,6]. Although the thermogenic role of BAT has been well established, BAT also acts as an endocrine organ to secrete cytokines and proteins (collectively termed batokines) to maintain nutritional homeostasis and regulate multiple-organ metabolism [7,8].

Physical exercise is an energy-consuming activity that activates lipid metabolism and promotes WAT browning [9]. Like cold stress, exercise is a non-pharmaceutical intervention to promote adipose tissue thermogenesis and browning. Since today’s society is generally faced with excessive energy intakes and physical inactivity, exercising to promote fat browning is considered to be an effective non-pharmaceutical intervention. However, the conclusions on exercise affecting adipose tissue browning and thermogenesis are inconsistent and even confusing in previous studies. In terms of the energy supply and demand, adipose tissue thermogenesis not only maintains body temperature, but compensates to sustain systemic energy balance, especially during acute and chronic exercise. Therefore, adipose tissue thermogenesis may play different roles in diverse physiological states. Previous studies indicate that the redox state controls the heterogeneity of adipose tissue thermogenesis at the cellular and molecular levels, thus leading to the conflicting conclusions that exercise has different effects on adipose tissue thermogenesis. This review aims to explain the contradictory results of exercise-induced fat thermogenesis and browning from several aspects and provide a novel interpretation: increased adipocytes’ browning and reduced BAT thermogenesis might be the terminals of exercise intervention on adipose tissue. Studies at different stages of the intervention process lead to conflicting conclusions. Finally, we focus on explaining UCP1-mediated fat thermogenesis heterogeneity controlled by a redox state.

## 2. BAT Biomarkers and Batokines

For humans and rodents, BAT differs from WAT by virtue of its small, multilocular lipid droplets and higher expression of special thermogenic proteins. UCP1, a highly expressed landmark protein of BAT, is located on the inner membrane of mitochondria and converts the proton gradient into heat by reducing ATP synthesis. Numerous studies have shown that induction of UCP1 increases thermogenesis in BAT and initiates the browning process in WAT [10]. UCP1 KO mice exhibited decreased thermogenesis upon cold stimulation [11] and dramatically increased body weight compared with the wild type during a high fat diet [12]. Moreover, recent studies have elucidated the mechanism by which ROS directly modifies UCP1 to stimulate thermogenesis [13]. Although emerging evidence indicates that UCP1-independent thermogenesis coexists in cells [14]. UCP1, which is the most crucial biomarker protein of BAT, is still the major contributor to physiological thermogenesis and fat browning.

Peroxisome proliferation-activated receptor γ coactivator-1α (PGC-1α) is another key regulator associated with thermogenesis. PGC-1α stimulates mitochondrial biogenesis and UCP1 expression, further promoting muscle fiber transformation (a switch to oxidative fibers) [15] and adipose tissue browning [16]. Depletion of PGC-1α led to reduced UCP1 and the failure of adipose tissue browning during the differentiation of mesenchymal stem cells (EMSCs), suggesting the essential role of PGC-1α in browning and thermogenic protein expression [17]. Moreover, the finding that the cannabinoid receptor (CB1) and UCP1 were both immunostaining as positive in interscapular BAT (iBAT), while both were negative in WAT, revealed a novel BAT biomarker [18]. Early B-cell factor 2 (Ebf2), pyruvate dehydrogenase kinase, isozyme 4 (Pdk4), heat shock protein B7 (Hsbp7) and F-box protein 31 (Fbxo31) are also highly expressed in human BAT [19]. In addition to the aforementioned BAT markers, previous studies have frequently identified PR domain-containing 16 (PRDM16), cell death inducing DNA fragmentation factor α subunit-like effector (Cidea) and Iodothyronine deiodinase 2 (DIO2), as emerging biomarkers of BAT and beige adipose tissue formation [20].

On the other hand, batokines secreted by BAT spurred a new wave of interest in their effects on modulating metabolism, such as fibroblast growth factor 21 (FGF21), bone morphogenetic protein 8B (BMP8b), interleukin-6 (IL-6), vascular endothelial growth factor A (VEGFA), insulin-like growth factor 1 (IGF-1), neuroregulatory protein 4 (NRG4), the lipokine 12,13-diHOME and microRNAs [21]. Exercise stimulates BAT to release batokines. For example, acute exercise enhanced the circulating lipokine 12,13-diHOME in humans and mice, which increased skeletal muscle fatty acid uptake and oxidation. Surgical removal of BAT decreased 12,13-diHOME levels [22]. Hepatic and plasma FGF21 were found at an elevated level after exercise, but the role of BAT-derived FGF21 in response to exercise remains to be confirmed. Likewise, while exercise increased VEGFA expression in WAT [23], exercise-induced VEGFA in BAT has not been reported. To our knowledge, few studies have focused on the effects of exercise-induced batokines. It is also unclear whether BAT is responsible for these changes upon exercise stimulation.

Although the aforementioned batokines are not BAT biomarkers, they make significant contributions to nutrient metabolism, immune system regulation and inflammatory response, serve as “crosstalk molecules” between BAT and other organs. Such crosstalk is likely to regulate energy expenditure and production across multiple systems and organs. While BAT landmark genes or batokines are also expressed in other tissues with high metabolic rates, the role of BAT in metabolic regulation has drawn great attention in recent years, and relative mechanisms can also be explained in these aspects [8,24]. BAT is not widely distributed in humans or animals, but the existence of these genes or factors does bring BAT into the metabolic regulation network of the whole body [25]. Figure 1 summarizes the main biomarkers of BAT and batokines that function between BAT and other organs.

## 3. Exercise-Induced Molecular Network Regulates Adipose Tissue Thermogenesis and Browning

### 3.1. Sympathetic Excitation

BAT and beige adipose tissue are innervated by the sympathetic nervous system SNS, which releases norepinephrine binding to β-adrenergic receptor (β-AR) to activate UCP1 and PGC-1α expression through the p38MAPK pathway [26]. Exercise enhanced the activity of ATGL and HSL by up-regulating β3 adrenergic receptors (β3-AR) in adipose tissue [27], activated BAT thermogenesis and induced the browning of WAT. β3-adrenergic stimulation elevated the expression of GPR81, a receptor of lactate, in adipose tissue accompanied by increased UCP1, which implied a potential lactate-mediated browning process [28]. The contribution of β3-AR to adipocytes’ browning is well established, but research indicates that other β-ARs are also involved in BAT thermogenesis. Blondin et al. illustrated that human BAT thermogenesis was mainly mediated by β2-AR [29], and the lack of β3-AR activity did not prevent this process in the subcutaneous WAT [30]. β1-AR KO mice exhibited cold-induced hypothermia and an attenuated response of BAT to catecholamine stimulation, suggesting the role of β1-AR in the SNS stimulation of thermogenesis [31]. In addition, the interplay between PRDM16 and zinc finger protein 516 (Zfp516) was found to modulate BAT activity and promote WAT browning via β-AR/cAMP signaling [32].

### 3.2. Muscle-Derived Cytokine Irisin

Irisin is an exercise-regulated cytokine secreted from multiple organs. Its precursor protein is fibronectin type III domain protein 5 (FNDC5), which is cleaved after translation to form mature irisin cytokines for release. Exercise-promoted myogenic irisin release, was found to further activate adipose tissue browning via Sirt1-dependent PPAR deacetylation [33]. Studies demonstrated that irisin-induced browning-related gene expression in WAT through the p38MAPK and extracellular signal-associated kinase (ERK) pathways [34], and beige adipocyte precursor cells, were more sensitive to irisin-induced browning [19]. During precursor cell differentiation, irisin inhibited adipogenesis while it stimulated bone formation [35]. Transcriptional activation cofactor PGC1-α, widely expressed in muscle, increased irisin to enhance UCP1 expression in subcutaneous WAT and energy expenditure [36]. Although the positive effect of irisin on adipose tissue browning has been reviewed extensively, several studies have expressed doubt whether exercise-induced irisin plays a role in this process. Human cells allowed irisin to selectively target a small subset of adipocytes in the classical BAT regions without affecting the major WAT depots [37]. Moreover, the temporary increase in irisin concentration after exercise makes it questionable whether irisin has a cumulative effect and can promote thermogenesis [38]. In the light of these reports, a consensus on the effect of exercise-induced irisin on adipose browning cannot yet be achieved [39].

### 3.3. Brain-Derived Neurotrophic Factor

Brain-derived neurotrophic factor (BDNF) has been proven as a key regulator for energy metabolism. BDNF contributes to energy metabolism and adipose tissue browning during exercise. Treatment with β3-adrenergic receptor agonist CL316,243 (CL) improved UCP1 and BNDF expression in the gonadal WAT of female mice, but decreasing the BNDF eliminated the induction of UCP1 by CL [40]. Acute stress (physical restraint stress for three hours) reduced skin temperature and body weight in mice. Through the irisin–BDNF pathway, BDNF could effectively inhibit the abnormal activity of hippocampal neurons and hypothermia caused by acute stress, elevate subcutaneous WAT thermogenesis and improve cognitive behavior [41]. Hypothalamic overexpression of BDNF elevated BAT markers (UCP1 and HSP60) in WAT and resulted in significant weight loss and fat depletion [42], suggesting a novel browning pathway mediated by the hypothalamic-adipocyte axis. Tyrosine Kinase receptor B (TrkB) is a BDNF-specific receptor encoded by the proto-oncogene TRK family and located on the nerve membrane. In addition to regulating cognitive function, impaired BDNF-TrKB receptor signaling has led to decreased energy expenditure and obesity [43]. Additionally, exercise also stimulated muscle-derived BDNF expression [44], thus BDNF and irisin from diverse tissues may act synergistically to promote BAT thermogenesis and adipocytes’ browning.

### 3.4. Fibroblast Growth Factor 21

Fibroblast growth factor 21 (FGF21) is a peptide hormone that is mainly expressed in the liver. FGF21 participates in a wide range of physiological activities, such as lipid metabolism, ketogenesis [45], insulin secretion [46] and the development of cancer [47]. Exercise is known to increase serum FGF21 levels in rodents and humans and improve adipose tissue sensitivity to FGF21 [48]. Treatment with FGF21 in vitro and in vivo both up-regulated the expression of browning genes in perirenal and inguinal WAT regions [49]. FGF21 promoted thermogenesis as well as PGC-1α expression in adipose tissue and skeletal muscle after exercise, via both the UCP1-dependent and -independent pathways [50]. Moreover, FGF21 induced adiponectin production by binding FGFR1c and β-Klotho, so adiponectin served as a downstream effector of FGF21 in white adipocytes and mediated the systemic effects of FGF21 on energy metabolism and insulin sensitivity in the liver and skeletal muscle [51]. With other myokines during exercise, irisin and FGF21 cooperated to stimulate browning [49,52]. However, the effects of different types and intensities of exercise on FGF21 are still controversial [53].

### 3.5. Metabolites

A number of metabolites are produced during exercise impacting fat thermogenesis and browning, such as FFA, creatine, acetate and succinate. Exercise increased lipolysis and released FFA that was bound to UCP1 and directly stimulated thermogenesis [54]. FFA also activated the nuclear receptor HNF4alpha, which subsequently promoted the liver to generate acylcarnitines as fuel for brown fat thermogenesis [55]. Creatine is enriched in muscle and adipose tissue. Muscle utilizes creatine metabolism such that mitochondrial ATP and creatine generate PCr and ADP in a 1:1 stoichiometry. During acute exercise, significant ATP is consumed to supply energy, thus the PCr pool drives substrate-level phosphorylation of ADP to synthesize ATP [56]. Arginine/creatine metabolism has been reported as a beige adipose signature [57]. Creatine has enhanced respiration and thermogenesis in beige fat mitochondria and has modulated this thermogenic mechanism in an UCP1-independent manner. Similar evidence was also reported by a recent study, the UCP1-deficient epididymal beige adipocytes employed creatine cycling as an alternate thermogenic mechanism [14]. Active skeletal muscle was able to enhance acetate release, uptake and oxidation [58]. Acetate treatment inhibited lipolysis, increased browning and thermogenic capacity in WAT of mice fed a high-fat diet [59]. In morbidly obese men, circulating acetate mediated the influence of gut microbiota composition on adipose tissue browning. The relative abundance of Firmicutes was positively associated with plasma acetate levels, which was linked to browning markers’ (PRDM16) expression in subcutaneous adipose tissue [60]. On the other hand, acute exercise increased succinate levels in adipose tissue both in mice and humans [61]. Succinate oxidation increased ROS production in adipose tissue which drove UCP1 to initiate thermogenesis [62]. In addition, β-aminoisobutyric acid (BAIBA) has been considered as one of the amino acid metabolites promoting adipocytes’ browning. Mouse and human studies demonstrated that exercise increased plasma BAIBA content [63,64]. BAIBA supplementation increased UCP1 and Cidea expression via the PPARα pathway, and accelerated fatty acid oxidation in the liver and thus reduced body fat [65]. Additional metabolites including γ-aminobutyric acid (GABA) and lactate were involved in adipocytes’ browning by modulating key thermogenic regulators and mitochondrial biogenesis [66,67], so as to maintain energy system homeostasis.

Collectively, BAT thermogenesis and adipose tissue browning may be adaptations to special physiological conditions, in order to sustain the balance between systematic energy supply and expenditure. Exercise regulates the browning and thermogenesis of adipose tissue through multiple organs (muscle, brain, liver and adipose tissue) and multiple pathways (nerve, endocrine and metabolic products), where the molecular mechanisms involved ultimately target thermogenic gene expression and mitochondrial biogenesis (Figure 2).

## 4. Adipose Tissue Browning during Exercise: Thermogenesis or Energy Waste?

Since the publication of the first study showing that exercise promoted WAT browning in mice, this conclusion has provoked a heated debate. Kizaki et al. found that browning and thermogenic gene expression elevated with increased BAT volume following a six-week swimming training in mice [68]. The up-regulated expression of browning-related genes in adipose tissue was detected after a one-week treadmill training in mice, accompanied by an increase in fatty acid transport and utilization [69]. Furthermore, a treadmill test with mice at 60% VO_2_ max intensity for six weeks showed that lipid droplets in the visceral WAT decreased markedly, and PGC-1 α/β mRNA levels were elevated with angiogenesis and mitochondrial biogenesis in BAT [70]. While studies confirmed the positive effect of exercise on activating adipose tissue browning and thermogenesis, some conclusions are just the opposite. BAT thermogenesis in endurance athletes was lower than that in sedentary people, and long-term endurance training was not associated with BAT activation and the recruitment of beige adipose tissue [71,72]. There was no difference in UCP1 expression between adipose tissues after exercise [73]. Similarly, Tsiloulis et al. indicated that beige adipose tissue formation in humans was not correlated with exercise, and the inconsistent conclusions between humans and rodents were just due to the distinct anatomical locations, molecular characteristics, and physiological functions of adipose tissue [74]. In Table 1 we compare the effects of acute [75,76,77,78] and chronic exercise [79,80,81] on adipose tissue thermogenesis and browning in animals and humans, and explain the possible molecular mechanisms involved (Figure 3).

Nevertheless, it is worth noting that many experiments with laboratory mice are usually done at 20–22 °C which is around the thermoneutral zone of clothed humans, but not the thermoneutral zone of mice (about 30 °C) [84,85]. Mice are actually in thermal stress under such conditions, which has a significant impact on the thermogenesis, energy expenditure, fat browning and other physiological activities [86,87]. Therefore, studies on mice living in standard housing conditions (20–22 °C) may not accurately reflect physiological changes in humans, who generally live in their thermoneutral zone. This may explain conflicting conclusions between studies on humans and rodents. Additionally, gender differences in BAT metabolism and WAT browning should be taken into consideration. Rodent and human studies illustrated that a few parameters influencing BAT metabolism and the browning process (such as adrenergic activation [88], UCP1 levels [89] and thermogenic gene expression [90]) demonstrated sexual dimorphism. PET/CT scanning data also showed a higher BAT activity in women than in men [91]. In women, preadipocytes from perirenal fat depots had the higher potential of browning [92]. These reports suggest that sex differences also lead to inconsistent results, which may be due to different estrogen and progesterone levels and the activation of adrenergic signaling pathways, or sex-specific intrinsic factors [88,92,93], but the predominant factor accounting for this phenomenon remains to be further confirmed. Intriguingly, although adipose tissue browning is often considered to alleviate obesity, the effect of exercise-induced browning varies among individuals with different degrees of obesity. For example, animal studies demonstrated that obesity blunted the response of WAT to exercise-induced browning [94] and obese rats were resistant to exercise-induced synthesis of mitochondrial and oxidative protein [95]. While exercise enhanced browning gene expression in individuals with different BMIs [9], it had no effect on obese men in another study [74]. Thus, the degree of obesity influences exercise-induced adipose tissue browning. This might be a consequence of impaired metabolic flexibility and signaling pathways, or attenuated adipose tissue sensitivity to exercise.

Importantly, it is impossible to equate the results of animal experiments with those in humans and to force the consistency of experimental results. In addition to the distinct thermoneutral zones we discussed above, there are physiological differences in the BAT response to exercise due to anatomical differences in BAT depots between animals and humans. As reported by Aldiss et al., mice at thermoneutrality demonstrated a muscle-like signature in BAT (see below) after exercise, but did not affect the thermogenic process [96]. Conversely, a positive association of habitual physical activity with thermoneutral BAT activity was reported in human subjects [91]. In addition, another animal study highlighted that the antioxidative role of UCP1 might establish prior to its thermogenic role during evolution, which is conducive to mitigating oxidative stress [82]. Therefore, comparisons between animal and human results need to be made with greater caution, and many of the current contradictions about exercise and fat browning may stem from the differences between species.

Why does exercise cause different results related to fat browning? Is it possible that fat browning serves a completely different role in humans from rodents? From the perspective of stress-response, increased BAT thermogenesis and adipose tissue browning show a trend of enhanced energy expenditure during exercise. However, increased thermogenesis results in a higher core body temperature. When the high core temperature is detrimental to exercise, BAT thermogenesis needs diminishing to decrease body temperature, or “cooling down”, as compensation to maintain normal physiological activities. Thus, increased thermogenesis stimulated by exercise may be a temporary phenomenon, captured by some experiments. From the perspective of energy supply and demand, the energy for BAT thermogenesis is supported by glucose and fatty acids oxidation, so thermogenesis may be an “energy waste” from the perspective of exercise as it is competing for energy with the skeletal muscle. Increased thermogenesis means that more blood and energy substrates are divided to adipose tissue, conflicting with the demand of the muscle for such substances. It is plausible that the body down-regulates the proportion of energy allocated to BAT in order to force more energy to go the right way—flowing into working the skeletal muscle. Evidence has underscored that the basal glucose intake was attenuated in brown adipocytes isolated from exercise-trained BAT compared with that differentiated from sedentary BAT [83]. Likewise, fatty acid oxidation in BAT was inhibited after exercise [97]. Therefore, the unchanged or even reduced heat production of BAT may be the “forced choice” during exercise, which has been captured by other experiments [71,98].

Moreover, there is another question pertaining to whether thermogenesis is the main aspect of BAT in response to exercise. With unaltered thermogenesis in obese mice after four weeks of swimming, BAT proteins (MYOD1, CKM and MYOG) involved in skeletal muscle physiological activities were upregulated [96]. Rosina et al. revealed that brown adipocytes under thermogenic stress released extracellular vesicles (EVs) that contained oxidatively damaged mitochondria into the macrophages to maintain the normal thermogenesis of BAT. Mitochondria-derived EVs impeded PPARγ signaling and UCP1 expression when re-uptaken by parental brown adipocytes [99]. Abundant mitochondria potentially increase the risk of mitochondria-derived EVs accumulation, thereby impairing thermogenesis. Therefore, exercise-induced mitochondrial biogenesis of BAT may not represent increased thermogenesis. Changes in BAT under exercise interventions perhaps contain some unknown physiological significance. The different perspectives and timings of observers may underlie the conflicting conclusions.

Here, we propose a possible explanation: BAT increases thermogenesis in the early stage of acute exercise and chronic exercise to consume energy and raise core temperature where an appropriate elevated body temperature is beneficial to promoting muscle strength, reducing the muscle viscosity and enhancing the activity of related enzymes, known as “warming up”, and finally accelerates the body to adapt to exercise. In the late stage of acute exercise with increased core temperature and depleted energy substrates, or in the late stage of chronic exercise training with the saving of energy substrates, BAT thermogenesis will be gradually lowered (Figure 4). This is consistent with the view of other authors that BAT activity will be turned down in response to a chronic higher core temperature [100]. It is interesting to note that these authors attribute the increased adipose tissue browning in other depots to a compensatory mechanism for decreased BAT function, given that rats showed a browning switch in subcutaneous depots, but this was accompanied with a reduced thermogenesis and a morphological transformation (whitening) in BAT following exercise [97].

Consequently, BAT thermogenesis and adipocytes’ browning cannot be simply defined as “exercise benefits”. Increased thermogenesis would be a temporary phase in response to exercise. Compared with the “inertia” of WAT, beige adipose tissue is a muscle-like intermediate phenotype with greater oxidative capacity, making it more responsive to exercise stimulation and more suitable for exercise. For all adipocytes, mitochondrial content determines the fat “color” and browning represents the enhanced oxidative capacity. Therefore, increased adipocytes’ browning and reduced BAT thermogenesis might be the most advantageous adaptation of adipocytes to exercise. Pontzer et al. found that the total daily energy expenditure (TEE) of people in primitive tribes who exercised intensely each day was about the same as that of sedentary people [101]. After conducting numerous metabolic studies, he concluded that: “regardless of lifestyle, there is a limit to how many calories the body burns each day, and the body seems to respond to increased daily exercise by using less energy for other tasks.” This implies that long-term exercise adaptation may cause a decrease in BAT thermogenesis. For sedentary people, the amount of energy used for exercise will be more evenly shared with BAT for thermogenesis and release, so as to achieve the “target” of daily energy expenditure. Overall, increased adipocytes’ browning and reduced BAT thermogenesis might be the terminal of exercise intervention in fat browning. Previous results diverge, perhaps because subjects are at different stages of the intervention process.

## 5. Redox Control of UCP1 and Thermogenesis

### 5.1. UCP-Mediated Thermogenesis

Many studies equate adipocytes thermogenesis with browning. In fact, browning reflects increased mitochondrial content in adipocytes, whereas thermogenesis reflects a process of oxidative phosphorylation uncoupling through UCP1. The primary thermogenesis in adipocytes is thought to be a process of UCP1-mediated”proton leak”. In general, the energy stored in the proton gradient is coupled with ATP synthesis. However, the energy is lost in the form of heat if it fails to synthesize the high-energy phosphate bonds of ATP, and UCP operates as the carrier for heat generation [102]. UCP removes the coupling relationship between electron transfer and phosphorylation in part of the normal respiratory chain, so that the oxidative phosphorylation process enters the idle state. Proton leakage mediated by UCP1 is regulated by fatty acids and purine nucleotides [103,104]. Fatty acids initiate proton leak via UCP1 through an H^+^/FA symport mechanism, whereas purine nucleotides block H^+^ transport by binding to the cytoplasmic surface of UCP1, thus preventing this process [54].

In addition to UCP1-mediated thermogenesis, other forms of UCP1-independent thermogenesis including the futile creatine cycle, calcium ions [105] and lipid circulation [106] also coexist in adipose tissue. The ATP/ADP carrier (AAC) had been suggested as contributing to cell thermogenesis. AAC mediated the proton leak sensitive to CAT (carboxyaluminate) upon activation by fatty acids, AMPs, or alkenals [107,108,109]. Brand et al. reported that AAC controlled half to two-thirds of the basal proton conductance of mitochondria. AAC1 knockout in mice halved the proton conductance of muscle mitochondria compared with the controls [110]. Moreover, creatine-mediated enhancement in respiration in beige fat mitochondria was abolished when AAC was inhibited [57]. An earlier study illustrated that UCPs and AACs in mitochondria improved proton transport following cold adaptation, resulting in heat production and increased cold resistance in king penguins [111]. Compensation pathways for adipose tissue thermogenesis have been revealed, that is, mitochondrial content does not determine the amount of heat production. Other UCP1-independent thermogenesis may also involve the whole energy metabolism. Regardless of what the proton transport carrier is, the proton gradient on the inner membrane of mitochondria must be the fundamental driving force for heat production.

### 5.2. Adipose Tissue Thermogenesis under Oxidative Stress

Many proton reflux channels are located on the inner membrane of mitochondria. Why choose UCPs for proton reflux and heat generation? What turns on the proton valve at the UCP? These are confusing questions. Reactive oxygen species (ROS) is produced by incomplete reduction of oxygen in the electron transport chain. Electrons that prematurely escape from the ETC generate superoxide anion subsequently transforming into hydrogen peroxide (H_2_O_2_) [112]. More than 10 sites within mitochondria are able to generate ROS [113,114]. ROS produced during exercise plays an important role in various physiological activities including regulating UCP1-mediated thermogenesis. The ROS and thiol oxidation state of adipocytes induced by physiology or drugs both increase cell thermogenesis. Mechanistically, ROS stimulates thermogenesis through the thiol oxidation of UCP1 at cystine (Cys253). Mitochondria-targeted antioxidant largely inhibited thermogenic respiration in BAT [13]. Superoxide dismutase 2 (SOD2) is a key enzyme to clear mitochondrial ROS. In SOD2 knockout mice, elevated ROS levels were accompanied by enhanced UCP1 expression and thermogenic energy expenditure [115]. Likewise, decreased SOD2 activity and redox imbalance were evident in adipose tissue from obese children [116]. These studies further illustrate the effect of ROS on promoting adipose thermogenesis. On the other hand, the regulation of thiol oxidation on fat browning and thermogenesis had also been confirmed in NRF2 knockout mice. NRF2 is a transcription factor that targets antioxidant enzymes and regulates ROS levels. Adipocytes showed higher thiol oxidation and WAT browning in NRF2 deficient mice [117]. Moreover, as a consequence of excessive intracellular oxidation levels, abundant accumulation of acetyl-CoA forced high acetylation of some key transcription factors (FoxO1), thus limiting their activity and impairing thermogenic and metabolic gene expression in adipose tissue [118,119]. A recent study revealed that Cys-253 was not essential for UCP1-mediated thermogenesis, but it improved the sensitivity of UCP1 to be activated by adrenergic stimulus [13]. Therefore, further investigations exploring other ROS-functional targets will be beneficial for us to gain a deeper understanding of ROS-regulated thermogenesis.

### 5.3. Adipose Tissue Thermogenesis under Reductive Stress

The redox milieu of a cell or tissue is facilitated by the reduction–oxidation potential of the redox couples (GSH/GSSG, NAD^+^/NADH and NADP^+^/NADPH, etc.). The abundance of either of these determines the cellular redox state: “oxidative” or “reductive” stress [120]. A more concise definition of reductive stress is the depletion of ROS below their physiological levels [121], which may be the consequence of increased reducing substance. Exercise produces both ROS and lactate that disturb the cellular redox state. ROS tends to compete for free electrons in lipids, proteins and DNA, leaving cells in a state of oxidative stress. Lactate is an intermediate product of glycolysis. In terms of the possibility of chemical reactions, it is an intracellular reducing agent. Lactate is oxidized and utilized under aerobic conditions and is also the main metabolite produced by skeletal muscle during exercise.

Lactate increased UCP1 expression in adipocytes and promoted WAT browning via the PPARγ pathway [122,123]. It induced the proliferation of brown/beige fat precursors through monocarboxylic acid transporters (MCTs) and upregulated key thermogenic markers including UCP1, PGC-1α and Cidea [124]. Piperine, a curcumin extract, enhanced lactate levels in myoblasts, which elevated UCP1 expression and BAT thermogenesis through AMPK signaling [125]. Moreover, obese mice had a lower level of GPR81 compared with the lean group and depletion of GPR81 largely diminished adipocytes’ browning and thermogenesis, a process that might occur via the p38MAPK signaling pathway [28]. Hypoxia-inducible factor (HIF) is a transcriptionally active nuclear protein regulating cellular response to stresses such as hypoxia, inflammation and tumor development. Wu et al. observed that intestinal HIF-2α mediated Ldha gene expression to enhance gut lactate levels, which activated the adipose G-protein-coupled bile acid receptor, GPBAR1 (TGR5). Activated TGR5 finally increased UCP1 expression and thermogenesis in WAT [126]. In the light of the above studies, lactate does act on adipose thermogenesis, but cells may be confronted with both reductive and oxidative stress in the context of exercise. Current evidence indicates that both ROS and lactate can stimulate fat browning and thermogenesis, as long as the delicate redox balance is disturbed, UCP1 will be induced. Thus, cell thermogenesis should be regarded as a necessary and common response to exercise, rather than a biomarker for health benefits (Figure 5).

## 6. BAT Thermogenesis Heterogeneity

### 6.1. BAT Heterogeneity

BAT is thought to originate in the mesoderm of embryonic development and shares a developmental origin with skeletal muscle, bone, WAT and connective tissue [127,128]. The specific developmental lineage of BAT is not fully elucidated. A previous model tends to suggest that interscapular BAT and skeletal muscle share a common precursor defined by Myf5 expression, but WAT is not part of this model [129]. However, emerging evidence indicates that many WATs also show as Myf5 positive, whereas some BATs are found to be negative [130,131]. Beige fat, on the other hand, is a type of adipose tissue between WAT and BAT that perhaps transforms from the precocious cell pool [132], or from existing adipose tissue [133]. In light of these reports, the adipocytes of different origin are distributed in the body. It is difficult to distinguish the function of adipose tissue from color alone. Therefore, it is worth noting that the molecular signature and metabolic phenotype are likely to display marked distinctions in heterogeneous adipose tissues. The adipocytes of adipose-specific insulin receptor knockout mice exhibited large and small polarization, which had different metabolic features and protein expression. Twenty-seven distinct proteins were detected between large and small adipocytes. The two types of adipocytes showed heterogeneity in lipid storage, glucose metabolism and regulators of cell metabolism [134]. In line with this, two subpopulations of brown adipocytes were found to coexist in mice, which were different in lipid droplet size, thermogenic capacity, mitochondrial content and morphology [135]. In addition, different fat depots can show completely opposite physiological activities in response to the same stress. For example, bone marrow adipose tissue does not undergo catabolism, but increases to some extent during a long-term caloric restriction diet [136]. Whereas a set of distinctions exist in morphology and gene expression, another study delineated an intertransformation between BATs upon a changed environmental temperature [135]. In general, adipose tissue heterogeneity is not only reflected in the developmental sources, morphology and gene selective expression, but also in the metabolic heterogeneity of adipose tissue at different locations. Lastly, the heterogeneity of mitochondria contained in beige adipocytes and BAT may reinforce the differences among adipose tissues [137].

### 6.2. Redox Control of BAT Thermogenesis Heterogeneity

BAT thermogenesis heterogeneity caused by different redox states is another possible mechanism to explain the contradictory conclusions under the same or similar exercise stress. It can be anticipated that the impacts of exercise are various among fat depots [83]. Radzinski et al. found a shifting ratio of cellular redox status in yeast cell populations over time, and cells showed distinct protein expression in different redox states [138]. This suggests that different ROS levels can lead to diverse physiological activities. That is, the thermogenesis controlled by ROS is a negative-feedback equilibrium: appropriate ROS levels stimulate thermogenesis, while excessive levels impair thermogenesis. Although the threshold for ROS to open the UCP1 proton channel is still unclear, given that UCP1 reduces the proton gradient via the “proton leak” process to decrease mitochondrial ROS release [139], opening the UCP1 proton channel is conducive to alleviating oxidative stress [140], which maintains heat production while preserving cell components. Consistent with these results, another study observed that UCP1 activity downregulated ROS in BAT mitochondria [82]. Adipose tissue thermogenesis synchronizes fatty acid oxidation with a transient increase of mitochondrial ROS, thus promoting the activation of redox-sensitive thermogenic signaling and the oxidation of UCP1 at Cys253. However, an overload of substrate flux to mitochondria causes a damaging ROS production that impairs mitochondrial flexibility [141]. The inflexible mitochondrial metabolism of adipocytes during obesity may correlate with the phenomenon that caloric restriction, which has been proven to promote WAT browning [142], fails to induce browning in the subcutaneous WAT of obese individuals [143]. Similarly, the damage of excessive ROS to BAT thermogenesis was also shown in thioredoxin-2 (TRX2) deficient mice, an antioxidative protein clearing mitochondrial ROS. The lack of TRX2 caused ROS overload and mitochondrial DNA release, which triggered the NLRP3 inflammasome pathway. However, the inhibition of NLRP3 rescued the decreased thermogenesis in BAT by ameliorating FA oxidation [144]. Consequently, increased intake of oxygen and mobilization of energy substrates during exercise possibly leave mitochondrial ROS production in a state of uncertainty: promoting thermogenesis or impairing mitochondrial flexibility. This leads to the BAT capacity of thermogenesis not being equivalent to the mitochondrial content. Furthermore, the distinct UCP1 expression within brown adipocytes [145] makes it more reasonable to speculate that ROS-mediated thermogenesis heterogeneity exists in BATs. Therefore, exercise-induced ROS in various BAT depots would result in thermogenesis heterogeneity because of its different levels.

Compared with oxidative damage, it is easy to overlook that a high level of reductive stress (glutathione over-accumulation) has provoked metabolic disorders in obese patients [146]. The appropriate decrease and oxidation of glutathione (GSH) elevates adipose tissue thermogenesis. Reducing GSH levels by buthionine sulfoximine (BSO) enhanced thermogenic gene expression (UCP1, Cidea and PGC-1α) and browning in WAT via activation of FoxO1 [147]. GSH depletion increased UCP2 and UCP3 expression in the adipose tissue of mice fed a high fat diet, thereby, up-regulating energy expenditure [148]. In addition, Mailloux et al. reported that a reductive cellular milieu would modulate BAT thermogenesis..A higher GSH/GSSG ratio (increased reductive stress) sensitized UCP1-mediated thermogenesis to GDP inhibition [149]. The depletion of Nrf2 in mice lowered GSH levels, but with higher ROS levels, which was accompanied by increased UCP1 expression in WAT [150].

Overall, the “battle” between oxidative stress (ROS) and reductive stress (lactate and GSH) is an important factor impacting adipocytes’ thermogenesis, but the underlying mechanism of how this “battle” is modulating adipocyte thermogenesis requires further elucidation and merits further investigation. In summary, BAT depots respond inconsistently to exercise [83] leading to different cellular redox stresses, which finally induce distinct UCP1 expression and BAT activation. Therefore, the various intracellular redox state of BAT causes thermogenesis heterogeneity.

## 7. Conclusions

Adipose tissue browning induced by exercise is the consequence of the interaction between ATP synthesis and heat production. Under the premise that the total energy is provided by the mitochondrial proton gradient, part of the energy is used for ATP synthesis and the other part for heat production. The differences in the cellular redox state during exercise can cause the thermogenesis heterogeneity of adipose tissue. The adipocytes’ heterogeneity is conducive to retaining energy homeostasis and metabolic flexibility in the face of complex stress, including exercise. Cells can determine the opening and expression of UCP1 by sensing the redox state, thus determining the proportion of energy consumption for heat production. However, the modulation of redox state on fat thermogenesis and browning warrants further mechanistic studies. Since additional energy loss is not favored for exercise, other physiological roles of exercise-induced adipose tissue browning and thermogenesis beyond energy metabolism will be interesting future avenues to pursue. As mentioned above, exercise-induced fat browning cannot simply be defined as reducing obesity or improving metabolism, and there should be various opinions and evaluations depending on physiological or pathological conditions. Particularly, some results with humans have been opposite to those for rodents, suggesting that exercise-induced adipose tissue browning should be treated with caution. Furthermore, the physiological activity that occurs in vivo in a living system does not occur in isolation, but is part of a metabolic pathway. In the context of exercise, investigating the connection between adipose tissue and other organs or systems may be instrumental in gaining a refined understanding of the benefits of exercise.

## Figures and Tables

**Figure 1 ijms-23-13142-f001:**
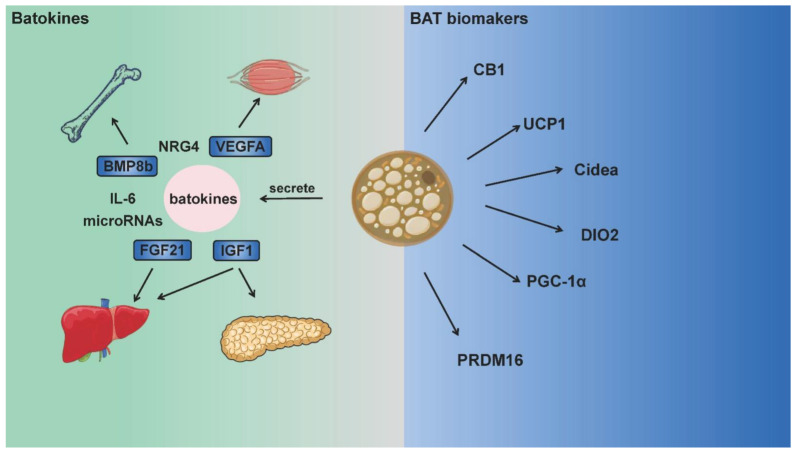
BAT biomarkers and batokines integrating energy metabolism between multiple organs. Brown adipose tissue is rich in mitochondria, and expresses UCP1, CB1, DIO2 and other thermogenic proteins as biomarkers. Moreover, batokines (such as FGF21, BMP8, IL-6) secreted by BAT act on multiple organs including the muscle, bone, brain, liver and pancreas. These batokines participate in numerous physiological activities as “crosstalk molecules” and integrate adipose tissue thermogenesis with the total energy metabolism.

**Figure 2 ijms-23-13142-f002:**
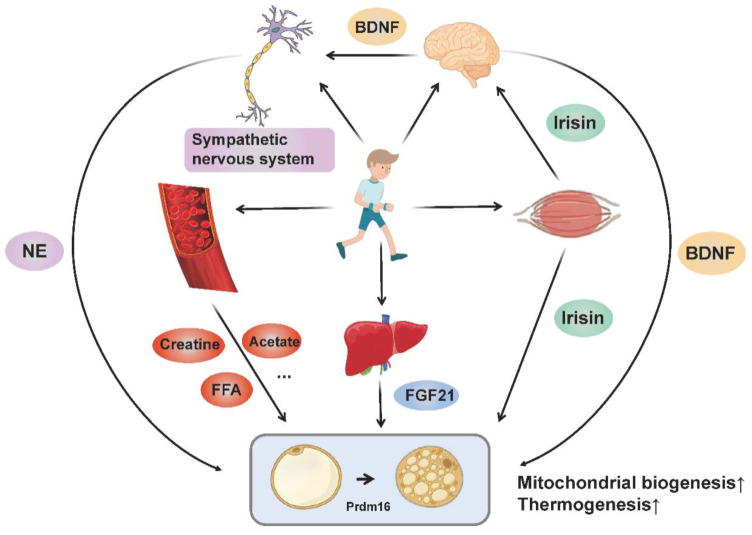
The molecular networks by which exercise promotes adipose tissue browning and thermogenesis. Exercise stimulates sympathetic excitation and releases norepinephrine binding to β-adrenergic receptors to promote adipose tissue thermogenesis and browning. In addition, exercise promotes the muscular release of irisin and hepatic release of FGF21. Irisin not only regulates the browning of adipocytes directly, but also promotes the release of BDNF from the brain through “muscle-brain crosstalk” and thus increases adipose tissue thermogenesis and browning. BDNF also acts on the sympathetic nervous system to mediate WAT browning. Circulating metabolites (such as creatine, acetate and FFA) produced by exercise also enhance adipose tissue thermogenesis and browning directly or indirectly. All these pathways ultimately target thermogenic gene expression and mitochondrial biogenesis.

**Figure 3 ijms-23-13142-f003:**
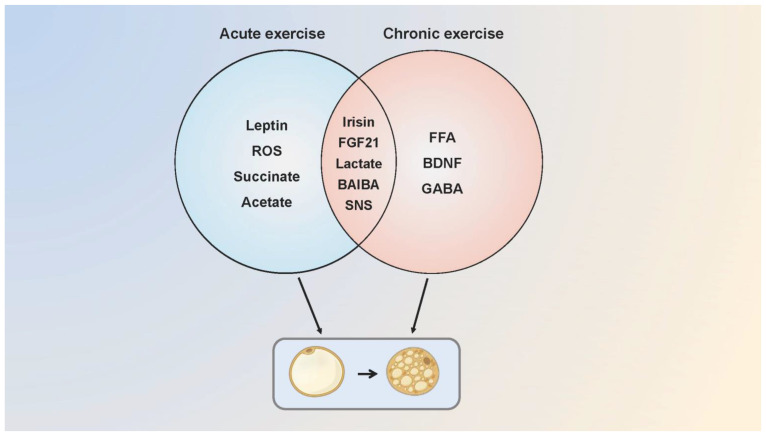
The possible molecular mechanisms of acute and chronic exercise that regulate adipose tissue browning. Acute and chronic exercise regulate the adipose tissue browning through a number of common and distinct molecular mechanisms. Acute exercise mediates browning through irisin, FGF21, leptin, lactate, ROS, succinate, SNS and acetate while chronic exercise possibly mediates through FGF21, SNS, FFA, BAIBA, BDNF, lactate and GABA. Some of the molecular mechanisms are speculative, as browning of adipose tissue has not been measured in the research. SNS, sympathetic nervous system.

**Figure 4 ijms-23-13142-f004:**
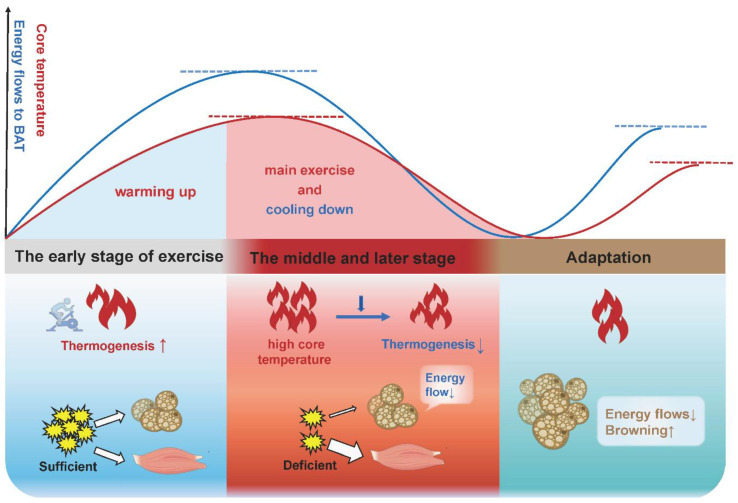
The physiological significance of adipose tissue thermogenesis and browning in different stages of exercise adaptation. In the early stage of acute exercise and long-term exercise, BAT increases thermogenesis to expend energy and raise the core temperature, which accelerates the body to adapt to exercise (warming up). In the late stage of acute exercise with an increased core temperature and depleted energy substrates, or in the late stage of long-term exercise training with the saving of energy substrates, BAT thermogenesis will be gradually lowered (cooling down). Beige adipose tissue is more responsive to exercise and more suitable for exercise, due to its greater oxidative capacity and more active metabolism. Thus, the increased adipose tissue browning is an adaptation upon exercise stimulation.

**Figure 5 ijms-23-13142-f005:**
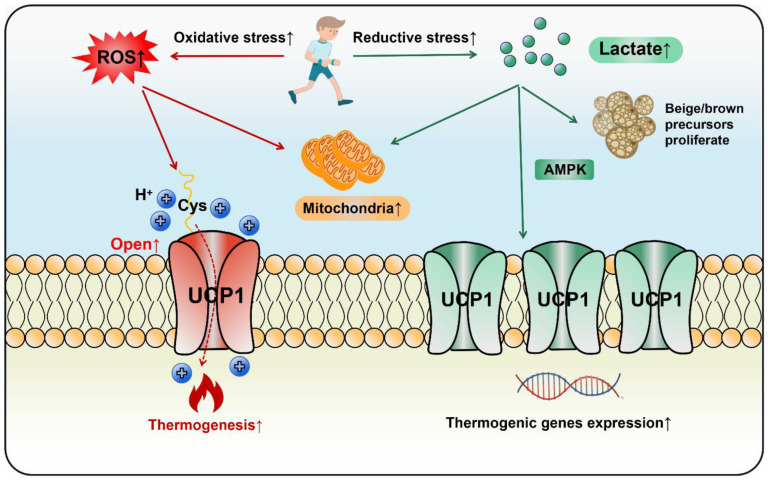
Both oxidative and reductive stress stimulate adipocytes’ thermogenesis and browning during exercise. In an oxidative stress state, ROS opens the UCP1 proton channel on the mitochondrial inner membrane to initiate thermogenesis by the thiol oxidation of UCP1 at cystine (Cys253). In a state of reductive stress, lactate as a reducing agent produced by exercise stimulates browning by up-regulating UCP1 expression through the AMPK pathway or directly promoting the proliferation of beige/brown precursor cells. In the exercise cycling of energy supply and expenditure (including one bout of exercise and chronic regular exercise), ROS initiates adipose thermogenesis immediately to compensate the daily energy expenditure, whereas lactate increases adipocytes mitochondrial biogenesis and browning through the adaptive up-regulation of UCP1.

**Table 1 ijms-23-13142-t001:** The effects of acute and chronic exercise interventions on adipose tissue thermogenesis and browning in humans and animals.

	Animal		Human	
	Acute Exercise	Chronic Exercise	Acute Exercise	Chronic Exercise
Adipose tissue browning	•UCP1 in subcutaneous WAT↑ [75] •C/EBPβ,PGC-1α, UCP1↑ [76] •FGF21↑ [77]	•PGC-1α and UCP1↑ [69] •FOXC2,multilocular adipocytes, UCP1↑ [70]	/	•UCP1,TBX1,CPT1B↑ [9] •FGF21,adipocytes sensitivity to FGF21↑ [48] •no effect [74]
BAT activity	Leptin, p-ERK1/2, UCP1↑ [78]	•mitochondria portein, UCP1, BAT volume↑ [68] •thermogenic response to NE↑ [79] •no effect [80,81] •PGC-1α,UCP1,fatty acid oxidation↓ [82] •oxygen consumption, in vitro thermogenesis↓ [83]	/	•unchanged browning markers, BAT activity↓ [71] •BAT volume and activity↓ [72]

↑ Indicators related to browning or BAT activity increase, ↓ Indicators decrease.

## Data Availability

Not applicable.

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
