# Peer review of "Exercise-Induced Adipose Tissue Thermogenesis and Browning: How to Explain the Conflicting Findings?"

_ijms, 2022, doi:10.3390/ijms232113142_

Round 1

Reviewer 1 Report (Previous Reviewer 1)

This review ranges over the evidence for the browning of white adipose tissue to form brown adipose tissue as a result of the impact of exercise and attempts to reconcile numerous contradictory findings.

Major criticism

This remains quite a controversial research area.  Indeed, there are many who are not wholly reconciled to the idea that brown adipocytes may persist into adulthood. The conclusions drawn by the authors of the current review are really very speculative in many instances.  There may be occasions where they have taken their speculation too far for a scientific publication.  One example of this concerns their concluding remarks regarding the switching of adipose tissue types in response to different types and intensities of exercise.  Although I can appreciate that this is a tempting speculation as a means of reconciling contradictory findings, it is not a convincing hypothesis due to the lack of evidence.  It could be that confining one’s speculations to those more closely based on evidence may carry greater impact.

Minor criticism

The English language used in the review really does require further attention, especially as regards the use of tenses and of plurals.  For example, the word “evidence” is normally used as a collective abstract noun meaning that it does not usually appear as a plural.

Author Response

We very appreciate the comments of the reviewers. We feel that the manuscript is now greatly improved. The following are responses to the reviewer's comments.

Point by point responses to the referee 1 comments

Major criticism

This remains quite a controversial research area. Indeed, there are many who are not wholly reconciled to the idea that brown adipocytes may persist into adulthood. The conclusions drawn by the authors of the current review are really very speculative in many instances.  There may be occasions where they have taken their speculation too far for a scientific publication.  One example of this concerns their concluding remarks regarding the switching of adipose tissue types in response to different types and intensities of exercise.  Although I can appreciate that this is a tempting speculation as a means of reconciling contradictory findings, it is not a convincing hypothesis due to the lack of evidence.  It could be that confining one’s speculations to those more closely based on evidence may carry greater impact.

Response: We completely understand the reviewer's concerns, thus we have removed the statement regarding "the switching of adipose tissue types in response to different types and intensities of exercise". We try to draw a conclusion from the current evidence to explain the conflicting results, one example is “increased adipocytes browning and reduced BAT thermogenesis may be the terminal of exercise intervention in fat browning.” As described in the article, from the perspective of energy allocation, it has been demonstrated that glucose uptake and fatty acid oxidation decreased in BAT after training. And the study also indicates that prolonged high core temperature will lead to a decrease in BAT thermogenesis. These provide some research support for our interpretation. Our current work has also been about addressing these concerns, setting up different exercise interventions to explore the true effects of exercise on adipose tissue thermogenesis and browning. We are trying to integrate the previous findings into the current work to make further efforts for our interpretation, hoping to have more convincing evidence to support our speculation. Therefore, we have modified the concluding statement in the paper.

Minor criticism

The English language used in the review really does require further attention, especially as regards the use of tenses and of plurals. For example, the word “evidence” is normally used as a collective abstract noun meaning that it does not usually appear as a plural.

Response: We have carefully revised the full text.

Reviewer 2 Report (New Reviewer)

Dear,

This is an important topic and the text of this manuscript is well written but at the moment MAJOR REVISIONS are necessary in order to make it suitable for a final decision for “IJMS”. In addition, there were topics similar to this study in the scientific databases as follows:

1- Exercise-Induced Adaptations to Adipose Tissue Thermogenesis

2- Exercise-induced ‘browning’ of adipose tissues

3- Exercise-Mediated Browning of White Adipose Tissue: Its Significance, Mechanism and Effectiveness

4- Endocrine Mechanisms Connecting Exercise to Brown Adipose Tissue Metabolism: a Human Perspective

POINTs of STRENGTH:

1) Review of mechanisms of adipose tissue thermogenesis and browning by exercise;

POINTs of WEAKNESS (and/or should be revised to improve the review manuscript):

2) Please specify the mechanisms of adipose tissue thermogenesis and browning by exercise, (mechanisms evaluated in this review) in the end of Abstract section;

3) In a very short sentence, please mention effective non-pharmacological interventions on adipose tissue thermogenesis and browning in the Introduction section;  

4) The purposes of this review article can be stated in more detail in the end of Introduction section.

5) Please explain the possible cellular and molecular mechanisms of the effects of short-term and long-term exercises on increased brown adipose tissue in schematic figures.

6) Please schematically explain the adipose tissue thermogenesis and browning by acute exercise (in three phases including warm-up, main exercise and cooling down/recovery [active and/or passive recovery]) and chronic exercise in type of exercises as well as different exercise intensities;

7) Please provide in the form of Tables previous studies “the effects of acute and chronic exercise interventions on adipose tissue thermogenesis and browning in humans and animals;

8) Please explain sex differences or different results for adipose tissue thermogenesis and browning by exercise in humans and animals;

9) Please explain the effects of exercise and BMI statuses (normal, overweight and obesity) on adipose tissue thermogenesis and browning; are there different results for this topic? Please clarify;

10) What are the conclusions and implications for future research?

11) What does this review add to the literature?

Best Regards

Author Response

We very appreciate the comments of the reviewers. We feel that the manuscript is now greatly improved. The following are responses to the reviewer's comments.

Point by point responses to the referee 2 comments

POINTs of STRENGTH:

1) Review of mechanisms of adipose tissue thermogenesis and browning by exercise;

POINTs of WEAKNESS (and/or should be revised to improve the review manuscript):

  • Please specify the mechanisms of adipose tissue thermogenesis and browning by exercise, (mechanisms evaluated in this review) in the end of Abstract section;

Response: We have added the mechanism in the abstract. We believe that fat thermogenesis and browning are a complementary regulation of energy metabolism during exercise, involving multiple organs and pathways and finally maintaining total energy balance in the body.

3) In a very short sentence, please mention effective non-pharmacological interventions on adipose tissue thermogenesis and browning in the Introduction section; 

Response: We have added this statement in the introduction. Like cold stress, exercise is a non-pharmaceutical intervention to promote adipose tissue thermogenesis and browning. Since today's society is generally faced with excessive energy intake and physical inactivity, exercising to promote fat browning is considered to be an effective non-pharmaceutical intervention.

  • The purposes of this review article can be stated in more detail in the end of Introduction section.

Response: According to reviewer’s comments, we added more detail statements in the article (see revised version).

  • Please explain the possible cellular and molecular mechanisms of the effects of short-term and long-term exercises on increased brown adipose tissue in schematic figures.

Response: We have added Figure 3 to explain this point .

6) Please schematically explain the adipose tissue thermogenesis and browning by acute exercise (in three phases including warm-up, main exercise and cooling down/recovery [active and/or passive recovery]) and chronic exercise in type of exercises as well as different exercise intensities;

Response: According to reviewer’s comments, we have made some modifications in Figure 4. Since BAT thermogenesis during exercise is a dynamic process and browning usually reflects the final result of exercise intervention, we describe the relationship between the energy flowing to BAT (represents thermogenesis to some extent) and the core temperature during exercise in Figure 4. Here, what is described in Figure 4 is a general case, dividing exercise (including acute and chronic exercises) into three basic phases: warming up—core temperature rise (energy abundance), main exercise and cooling down (energy deficiency) and adaptation (decreased thermogenesis, increased browning). The reason why we integrate acute and chronic exercise into one exercise model is that the three basic phases are similar in the two types of exercise.

7) Please provide in the form of Tables previous studies “the effects of acute and chronic exercise interventions on adipose tissue thermogenesis and browning in humans and animals;

Response: We have added Table 1 to explain this point.

  • Please explain sex differences or different results for adipose tissue thermogenesis and browning by exercise in humans and animals;

Response: We have added some sentences in chapter 3 to explain this point (see revised version).

  • Please explain the effects of exercise and BMI statuses (normal, overweight and obesity) on adipose tissue thermogenesis and browning; are there different results for this topic? Please clarify;

Response: Although many experiments on exercise-induced fat browning are conducted on obese individuals to reduce obesity, the degree of obesity has an impact on the effects of exercise-induced browning. Obesity blunted the response of WAT to exercise-induced browning in mice [93] and obese rats were resistant to exercise-induced synthesis of mitochondrial and oxidative protein [94]. In humans, while exercise enhanced browning gene expression in individuals with different BMI [9], it had no effect in obese men in another study [75] (see revised version).

  • What are the conclusions and implications for future research?

Response: We added some related descriptions in the conclusion.

  • What does this review add to the literature?

Response: This review attempts to present some conflicting results from previous studies and to present a possible explanation from the perspective of integrated physiology and total energy balance. Explaining these contradictions will help prevent similar research from going nowhere in the future.

Round 2

Reviewer 2 Report (New Reviewer)

Dear,

Manuscript Number: ijms-1981664

Title Manuscript: Exercise-induced adipose tissue thermogenesis and browning: How to explain the conflicting findings?

Thank you very much for the efforts of the authors.

In general, this manuscript has found suitable content after correcting major revisions, and the modified revisions are accepted.

Best Regards

25 October 2022

This manuscript is a resubmission of an earlier submission. The following is a list of the peer review reports and author responses from that submission.

Round 1

Reviewer 1 Report

The authors review some of the recent studies describing the effects of exercise and exercise training on the activity of brown adipose tissue.  One of their stated intentions is to reconcile some of the conflicting evidence emergent from research based publications.

Major comments

One of the fundamentally confusing issues surrounding research on brown adipose tissue and one which has confounded many interpretations of findings is the difficulty in reconciling outcomes of studies on experimental animals with those on human subjects.  There are fundamental differences in the occurrence and location of BAT depots in humans and animals which should be addressed in any review of this material.  Throughout the current review, the authors seem to have assumed that BAT deposition in humans is similar to that in experimental animals.  This has led them to freely translate the results of animal studies to the human condition.  This really cannot be done and the authors should include an appropriate discussion of the difficulties in drawing conclusions about human BAT from animal studies.  They should also address the compatibility of human and animal findings at each stage where they have chosen to cite relevant experiments.

The section on UCP homologues is superficial to the stage of being redundant.  There are many UCP homologues of wide tissue distribution but, other than UCP1, probably most do not function as uncouplers.  There has been a great deal of recent research aimed at elucidating roles of different UCPs and the emerging situation is fairly complex.  The UCP section in the current review must be either updated to include meaningful observations regarding UCPs and their roles in BAT and in whole body energy coordination or it should be removed.

Minor comments

The authors have embarked on the policy of citing reviews as original sources for statements on which they rely for their arguments.  In my opinion, this is a poor practice.  Original references should be cited when stating findings.

The review is written in such a way as to clearly convey meaning in most instances. However, the standard of English language usage falls some way below what would be expected for an international journal.  There are many examples of misuse.  I would recommend that the manuscript be edited by a native English speaker.

Reviewer 2 Report

This article review discusses the effect of physical exercise on adipose tissue thermogenesis and browning and the correlation with REDOX level of cells determined by the levels of lactate and ROS produced during physical exercise. The heterogeneity of intracellular REDOX state induced by exercise gave divergent results in previous studies. The role of the mitochondrial

 UCP1 proton channel is also discussed. The review should clarify “the conclusions on exercise affecting adipose tissue browning and thermogenesis are inconsistent and even confusing in previous studies”.

Just reading the title and summary the aim of the paper is not clear: the title talks about “REDOX control of exercise” and the abstract talks about “Lactate and ROS production during exercise may determine the REDOX level of cells.”; this central point of the article is not described clearly enough. In addition in the text both “REDOX” and “redox” are both present on line 22 and at the end, lines 24-25 “possible mechanisms of UCP1-mediated fat thermogenesis heterogeneity controlled by redox.” not followed by level  or states  or control. it is important that each word / abbreviation is always reported the same way and with the same meaning.

Line 31 “BAT is rich in uncoupling protein 1” should be better “BAT is rich in mitochondria that contain uncoupling protein 1”

Line 31 “proton motive force to ... “ should be better proton motive force for ATP synthesis to..”

stimuli at Line 36 (cold, exercise, β-adrenergic agonists) generate, in adults, from WAT a brown-like adipose tissue, named beige adipose tissue; this phenomenon represents browning. Elsewhere in the text, es Line 41-42 the authors write that physical activity promotes transformation of WAT to BAT, this point is controversial: beige and BAT are used in the text with the same meaning, one should indicate browning as a WAT to beige transition or as a WAT to BAT transition.

From line 30 to 35 there are no references

Line 58-59 “PGC-1α promotes muscle fiber transformation”, what does it exactly mean? Several sentences are not clear, for example how UCP1 promotes thermogenesis?

Line 63 different not explained proteins/factors are listed

Lines 93-95 no reference is present for the sentence.

Line 121 .... hypothermia caused by acute stress, this sentence too is inserted into the text and not explained

Lines 131-132 FGF21... delays the development of tumors; the relevant sentence needs a reference and more details

Lines 134-135 ..., as well as the increases expression.. it is not written correctly

Line 136 β-Klothoinduce, WHATdoes this word mean? Do you mean coreceptor Klotho?

Lines 219-220 and following “BAT mitochondria released extracellular vehicles (EVs) containing oxidatively damaged components into the macrophages” this sentence is unclear and must be rewritten correctly, the role of mitochondrial derived EV is not well explained and is not clear when these vesicles are released.

Line 225 and Fig 3 “Here, we propose a novel understanding”; this is a review, the authors must report and comment published data not to propose hypotheses that cannot be proved by experimental data, only an interpretation can be given.

Line 225-226 What does it mean that in the early stage of exercise the  thermogenesis in BAT accelerates the body to adapt to exercise?

Lines 245-246 “Compared with the "inertia" of WAT, beige adipose tissue is a muscle-like intermediate phenotype with greater oxidative capacity, making it more responsive to  exercise stimulation and more suitable for exercise.” the sentence makes a very innovative / prominent statement; is it supported by papers/evidences?

Line 262 UCP instead of UCP1

Lines 276-277 The following sentence is not clear “Compensation pathways for adipose tissue thermogenesis have been revealed, that is, mitochondrial content does not determine the amount of heat production”

Lines 278-279 the conclusion “It's dictated by the basic laws of thermodynamics.” Is not explained

Line 300 is instead of are

Lines 305-306 Is the definition of “thermogenic respiration” correct?

Lines 309-311 the following sentence is not clear in my opinion “From the point of view of obese children, fat thermogenesis is increased, which does not prevent them from becoming obese children. It can be judged that in this case, fat thermogenesis is a compensatory energy consumption of obesity.”

Line 315 this sentence should be rewritten “when intracellular oxidation level is excess”

Line 327 piperine is present without any explanation, for example one can write: piperine, a curcumin extract...

Line 330 the following sentence may be reformulated “by which the mechanism might through lactate-GPR81-p38MAPK pathway”

Lines 335-336 The concept of reductive stress it is not well known to most readers and would need more explanation in the text; the sentence “but cells must face both reductive stresses and oxidative stress in the context of  exercise” does not have sufficient basis. The same goes for the closing sentence “Thus, cell thermogenesis should be regarded as a necessary and common response to exercise, rather than a biomarker for health benefits”

Line370-371 “under special conditions” used in the text is extremely vague and should be substituted with a more precise indication such as “changes in environmental temperature”

Line 381 “oxidized and reduced cells” should be better described by “cellular redox status....”

Lines 382-383 the sentence “That is, the thermogenesis controlled by ROS is a negative feedback equilibrium per se.” needs to be rephrased and explained  better

Although the review offers some interesting observations  (es Lines 194-195) concerning the criticism on previous experiments:  “studies on mice living in standard housing conditions (20-22℃) may not accurately reflect physiological changes in humans, who generally live in their thermoneutral zone. This may explain the conflict of conclusions between humans  and rodents.” And also interesting observations on gender differences, however the attempt to clarify the discrepancy of results concerning the role of exercise as a way to get WAT browning and thermogenesis or energy waste is not exhaustive.

In conclusion although the topic of the review is interesting, it does not provide a sufficient clarification to what is already present in the literature.

This may also be partly due to incorrect English which often gives rise to ambiguous statements
